# The Effect of Menopause Hypoestrogenism on Osteogenic Differentiation of Periodontal Ligament Cells (PDLC) and Stem Cells (PDLCs): A Systematic Review

**DOI:** 10.3390/healthcare9050572

**Published:** 2021-05-12

**Authors:** Edoardo Di Naro, Matteo Loverro, Ilaria Converti, Maria Teresa Loverro, Elisabetta Ferrara, Biagio Rapone

**Affiliations:** 1Interdisciplinary Department of Medicine, University of Bari, 70121 Bari, Italy; edoardo.dinaro@uniba.it (E.D.N.); matteo.loverro01@icatt.it (M.L.); m.loverro2@gmail.com (M.T.L.); 2Department of Emergency and Organ Transplantation, Division of Plastic and Reconstructive Surgery, “Aldo Moro” University of Bari, 70121 Bari, Italy; ilaria.converti@gmail.com; 3Complex Operative Unit of Odontostomatology, Hospital S.S. Annunziata, 66100 Chieti, Italy; igieneeprevenzione@gmail.com; 4Department of Basic Medical Sciences, Neurosciences and Sense Organs, “Aldo Moro” University of Bari, 70121 Bari, Italy

**Keywords:** Menopause, oestrogen, oestrogen deficiency, oestrogen receptors, periodontal disease, periodontitis, periodontal ligament cells

## Abstract

(1) Background: Menopause is a physiological condition typified by drastic hormonal changes, and the effects of this transition have long-term significant clinical implications on the general health, including symptoms or physical changes. In menopausal women, the periodontium can be affected directly or through neural mechanism by oestrogen (E2) deficiency. The majority of the biological effects of E2 are modulated via both oestrogen receptor-α (ERα) and oestrogen receptor- β (ERβ). There is evidence that hypoestrogenism has a substantial impact on the aetiology, manifestation and severity of periodontitis, via the regulation of the expression of osteoprogesterin and RANKL in human periodontal ligament cells through ERβ. However, the mechanistic understanding of oestrogen in periodontal status has been partially clarified. The aim of this paper was to synopsize the recent scientific evidence concerning the link between the menopause and periodontitis, through the investigation of physio-pathological impact of the oestrogen deficiency on osteogenic differentiation of PDLSCs and PDLSC, as well as the dynamic change of ERα and ERβ. (2) Methods: Search was conducted for significant studies by exploring electronic PubMed and EMBASE databases, and it was independently performed by two researchers. All studies on the impact of oestrogen level on alveolar bone resorption were searched from 2005 to July 2020. Data selection was in concordance with PRISMA guidelines. (3) Results: Eight studies met the criteria and were included in this systematic review. All studies reported that oestrogen deficiency impairs the osteogenic and osteoblastic differentiation of PDL cells and oestrogen affects the bone formation capacity of cells. Seven studies were conducted on animal samples, divided into two groups: the OVX animals and animals who received the sham operation. (4) Conclusions: There is a multitude of data available showing the influence of menopause on periodontal status. However, the evidence of this line to investigation needs more research and could help explain the physiological linkage between menopause state and periodontal disease.

## 1. Introduction

The incidence and prevalence of periodontal disorders are subject to several factors, including the hormonal fluctuation [1,2,3]. Insufficient oestrogens release may have a greater impact on periodontal tissues of women during menopause or already suffering from pre-existing periodontal disease [2,3,4]. Menopause is the result of loss of ovarian follicular function and coincides with spontaneous cessation of menstrual cycle. This period is typified endocrinologically by the decreasing of ovarian activity with a substantial change occurring in the source and nature of circulating oestrogens [1,2,3,4]. The progressive oestrogen decline, which characterized the menopause, is also a driving force behind the reproductive aging (RA), a time-dependent deterioration that involves the female germ cells, which occurs in women older than 35 years old [1,2,3,4]. Periodontal destruction is manifested clinically by progressive changes in the architecture of the periodontal structures. The integrity of the periodontal tissues is regulated by several factors, including the capability of periodontal ligament of differentiating into osteoblasts or cementoblasts [5]. It has been emphasized that dietary supplement, nutraceutic and anabolic drugs could influence periodontal bone health in this population. It has been established the crucial function of the normal vitamin D status in skeletal mineralization, pointing out on its role in immunity and inflammatory pathway which is revealed through an increased release of anti-inflammatory cytokines and decreasing pro-inflammatory cascade [5,6].

Moreover, the characteristics of oestrogens deficiency include the induction of anthropogenic substances activity, distinguished by estrogenic properties. Among them, have been found a group of exogenous substance identified as endocrine disruptors to leach endocrine-disrupting chemicals, that interfere with bone remodelling process and calcium metabolism, exerting deleterious and pleiotropic effects on reproductive and bone tissues [6].

Investigations concerning the oral microcirculation in postmenopausal women have revealed that the diameter of the loops, tortuosity of vessels in labial mucosa and density of periodontal mucosa are altered and predispose to periodontal inflammation. The swelling of endothelial cells and periocytes of the venules, adherence of granulocytes and platelets to vessel walls, increased vascular permeability and vascular proliferation have been associated with variations in the oestrogen levels [6,7,8]. The suppressed oestrogen activity is the suggested major common cause of the decrease of alveolar bone remodelling [1,2,3]. During the menopause, the decline in ovarian oestrogen production disrupts the hypothalamic-pituitary-ovarian axis, leading to physiologic changes, which involve the bone in a determinant way [9,10,11]. Typically, osteopenia characterizes the menopausal status [12]. The osteoporosis, an oestrogen deficiency condition, is the most prevalent disease in menopausal women. Besides the increased risk of osteoporosis and fractures, the dynamic process that involves the bone metabolism and remodelling jeopardise the health of periodontium [1,5,13]. Indeed, the balance between the number and activity of osteoblasts and osteoclasts carry out this cycle. In vivo and in vitro studies demonstrated that disruption of the positive feedback on the hypothalamic-pituitary-ovarian axis leads to development of alveolar bone resorption [4,7]. One of the major events that cause the progressive architectural disruption of the periodontium during menopause is the impairment of the normal alveolar bone remodelling cycle, via the excessive release of the cytokines among them RANK ligand (RANKL—nuclear factor-kappa B ligand) by osteoblasts, which are directly implicated in the osteoclastogenesis cascade. Oestrogen deficiency leads to an intensification of the immune response due to the deregulation of T cell function and immune cell bone interactions [8,11], which leads to an increased T cell production of tumour necrosis factor (TNF). RANK, a member of the tumour necrosis factor (TNF) ligand family expressed on osteoblast/stromal cell membranes, plays an important role in coupling disproportionate increase in bone resorption [2,14]. The oestrogen deficiency is also responsible for the simultaneous decrease in osteoprotegerin (OPG) secretion, a natural decoy receptor against RANKL [1,15,16,17]. The OPG/receptor activator of NF-κB ligand (RANKL) system is one of the major downstream mediators of the action of oestrogen on bone [8,9]. Both processes, as well as the physiological reduction of skeletal mechanical stimulation, lead to the fall of bone strength. More recently, some authors also gave an outline of the actions of oestrogens, focusing on the physiological effects on the periodontal ligament cells of animals and human cell lines [3,16]. More specifically, some studies have explored the role of oestrogen in the potential for osteogenic differentiation of periodontal ligament stem cells (PDLSCs), periodontal ligament cells (PDLSC), human periodontal ligament stem cells (hPDLSCs) and periodontal ligament cells (hPDLSC), which indicate the crucial impact of hypoestrogenism on alveolar bone resorption during menopause. In human periodontal ligament cells, a reduced collagen synthesis in fibroblasts has been reported in physiological concentration of oestrogens [10]. Consequently, the gradual decrease of oestrogen levels induces a dose-dependent intensification in the production of pro-collagen I [5]. Oestrogen modulates the activity of target cells by binding specific intracellular oestrogen receptors (ER): Erα and Erβ [13,14,15,16,17]. Previous studies have indicated that both Erα and Erβ were expressed in animal bone marrow mesenchymal stem cells (BMSCs) and human PDLCs [3,8]. These hormone receptors mediate the regulation of cell growth and differentiation in response to oestrogens. The biological action of ERα is determined by on its constitutively active AF-1 and ligand-dependent AF-2 domains, while ERβ depends only on its AF-2 domain [4,9,15]. ERα and ERβ either homodimerize or heterodimerize upon ligand binding, and then regulate transcription of downstream target genes by interacting with either coactivator. Real-time PCR analysis have shown the lower expression levels of oestrogen receptors (ERs: ERα and ERβ) mRNAs in PDLCs [18,19,20]. These results denoted a crucial role of Erβ in oestrogen-induced effects on osteoblastic differentiation function of PDL cells and the oestrogen influence on the bone formation capacity of PDL cells through Er β [1,13,16].

The impact of oestrogen deficiency state of menopause has gained growing attention. In this setting, this review was aimed to discuss the dynamic correlation between physiological changes during the menopause and the consequences on the alveolar bone microenvironment.

## 2. Materials and Methods

### 2.1. Search Strategy

A systematic review was performed using the PRISMA statement recommendations (Figure 1) [21].

Searches were conducted in PubMed and Embase (through Ovid) (2005–2020) to identify relevant studies which investigated the relationship between menopause and periodontal status, and this search was independently performed by two researchers. The electronic searches were supplemented by manual searching of reference lists and reviews identify additional primary studies. The complete search strategy for each database is available in Table 1.

Two researchers separately searched for articles using the following terms: (menopause) OR (menopausal status) AND (oestrogen) AND (oestrogen receptor) AND (oestrogen receptor-α) AND (oestrogen receptor-β) AND (periodontal disease) OR (periodontitis) AND (bone mineral content) OR (bone mineral contents) OR (osseous density) OR (bone density) AND (RANK Ligand) OR (RANKL) AND (RANK) OR (RANK receptor) AND (osteoprotegerin) AND (periodontal ligament stem cells (PDLSCs)) AND (periodontal ligament cells (PDLSC)) AND (human periodontal ligament stem cells (hPDLSCs) AND periodontal ligament cells (hPDLSC).

### 2.2. Study Selection

Studies that met the following criteria were included in the review: human and animal studies which focused on the impact of oestrogen levels on osteogenic differentiation of Periodontal Ligament cells (PDLC) and Stem Cells (PDLCs) of animal and human cell lines, and studies exclusion criteria were as follows: (1) reviews, comments and letters; (2) abstracts or thesis publications; (3) duplicated publications. 

### 2.3. Data Extraction and Quality Critical Evaluation

Two investigators performed the study selection, data extraction and quality independently, and disagreements were resolved by discussions. The extracted information included the study type, first author’s name, publication year, characteristics of the study, and outcomes information. The risk of bias assessment tool was used to assess the quality of randomised controlled trials, by two authors separately.

## 3. Results

### 3.1. Summary of the Data Retrieval

In total, the citation screening for the review on menopause age and periodontitis produced 115 manuscripts from PubMed and EMBASE. At the initial screening carried out for developing the comprehensive bibliographic literature database, 115 articles were identified from the sources. After excluding duplicates, the selected publications were scrutinized separately by two reviewers. One hundred papers were excluded based on the title and abstract. Then, 8 articles were finalized, as highlighted in Table 1. All retrieved citation records were screened on the basis of eligibility criteria. Of all the included studies, 2 [25,29] were animal studies, while 6 were in vitro human cell studies [22,30]. The quantitative analysis cannot be conducted because of the heterogeneity in designing experiments, the discrepancy in conducting quantitative research study, the differences in studying the biological characteristics of species across the studies. Two studies investigated hPDL cells. A total of three strains were used in these studies, including ddy, Sprague-Dawley, Swiss-Webster, mice and rats. The OVX was performed about 4 to 25 weeks of age. All studies used post-OVX periods, and in the postoperative period follow up examinations were performed at 3–4 weeks to 3 months. With regard to the sample size, no trials reported any sample size computation. The median sample size of the OVX and control groups in all 7 studies was 23 ± 7. The main characteristics of the eligible studies are shown in Table 1.

### 3.2. Risk of Bias Assessment

The risk of bias for all studies is shown in Table 2. 

Total scores ranged from 1 to 4 (max = 4). A quality score of 4 was assigned to 2 studies [26,27]. Only one study was awarded a single point [28], and two awarded two points. Of all studies, two were awarded the highest quality score. No blinding method was classified. Six studies stated the random allocation of samples to groups, but the method of randomization was not explained [23,24,25,26,27,29].

## 4. Discussion

The majority of studies observed the microarchitecture deterioration in the alveolar bone that characterizes the osteoporosis. The association between the post-OVX period and Bone Mass Density (BMD) changes has been evaluated in several studies. A previous meta-analysis described a more notable decrease in BMD in animals with longer post-OVX periods. This systematic review identified publication bias for the outcomes of ERβ. In recent years, and especially in the last decade, the phenomenon of the relationship between periodontitis and systemic health is attracting a lot of attention around the research. A multidimensional paradigm is based on the view that periodontal inflammation could activate physiological changes and responses and, conversely, how systemic disorders may affect the periodontal attachment apparatus. It has been amply recognized that the incidence of periodontal disease increases rapidly remarked in menopausal women, and that the severity of periodontal tissue deterioration is more evident. The course of menopause is dictated by the ovarian function decline, accompanied by a substantial reduction in circulating sex steroid hormones. The in vitro study conducted by Mariotti A.J. [18], expanded previous research to reveal the importance and role of oestrogen in the proliferation of gingival fibroblasts. The author shown that premenopausal fibroblasts cells incubation, depending on oestrogen enrichment, in presence of 1nM oestradiol, lead to substantial changes in cells proliferation. This study illustrated that fibroblasts cells proliferation is determined also by the circulating levels of oestrogen, consistent with other studies. Several studies focused on the association between the hypoestrogenism and the bone mass density (BMD). Many authors used a mouse model to study the downstream effects of hypoestrogenism in inducing osteoclastic bone resorption. They determined the effect of oestrogen deficiency on the alveolar bone mass of OVX female mice and demonstrated a markedly decrease of this hormone after the operation, showing a significant difference between the OVX and sham-operated groups. The connection between the oestrogen fluctuation and pathophysiology of periodontal disease has gained great interest for the past decade. However, the biological plausibility of this association has been partially understood, especially regarding the impact of the oestrogens on periodontal ligament cells. Several studies have revealed that decreased oestrogen levels in menopausal women are related to morphological and inflammatory response of periodontal tissues, and there is a growing evidence that periodontal changes are influenced by the definitive cycle of oestrogen secretion. Therefore, we performed a systematic review the data acquired from concerned studies to explore the potential association between the oestrogen deficiency in menopausal status and the potential of osteogenic differentiation of periodontal ligament cells and stem cell line. These two phenomena take place simultaneously and are interconnected through a complex network of interactions and feedback regulations. As a steroid hormone, oestrogen regulates many physiologic processes, including the maintenance of the skeletal homeostasis by regulating the bone mineral density. Bone remodelling reflects a balance between the osteoclasts and osteoblasts. In vivo and in vitro studies demonstrated that oestrogens release inhibits the expression of osteoclastogenic factors for osteoclastogenesis [2]. Furthermore, oestrogens exert a protective effect by promoting the apoptosis of osteoclasts via regulation of cytokines expression, including interleukin-1, interleukin-6, receptor activator of nuclear factor kappa-B ligand (RANKL), tumour necrosis factor-*α* (TNF). Several preclinical and clinical studies concerning the effect of E2 on the bone mesenchymal stem cell (BMSC) have shown to enhance bone proliferation and osteogenic differentiation in various species, including mouse, rat, and human [8,9,10]. However, the exact molecular mechanism underlying the observed effects of E2 on osteoblast differentiation has not yet been fully understood. The role of oestrogen deficiency as a developer of postmenopausal periodontitis on human has been based on the observation about osteoporosis. The link, materialized in its connection to increased bone metabolism, is mediated by essential non-redundant factors of osteoclast biology: RANKL, RANK, and osteoprotegerin (OPG)/receptor activator of NF-κB ligand (RANKL) system. RANKL is a cytokine which promotes the proliferation and survival of osteoclasts by binding its receptor RANK. Oestrogen deficiency leads to an increase in the immune function, which culminates in an increased production of TNF by activated T cells. TNF increases osteoclasts formation and bone resorption both directly and by augmenting the sensitivity of maturing osteoclasts to the essential osteoclastogenic factor RANKL. Increased T cell production of TNF is induced by oestrogen deficiency via a complex mechanism mediated by antigen presenting cells and involving the cytokines IFN-g, and TGF-b. Experimental evidence suggest that oestrogen prevents bone loss by regulating T cell function and immune cell bone interaction. The present review was conducted to examine the current evidence about the correlation between menopausal status and periodontitis. Oestrogen has been shown to have anti-inflammatory, anticancer and immune-regulatory effects, in addition to its traditional role in regulating calcium and phosphorus metabolism [30]. It has been reported that oestrogen deficiency is commonly observed in jaw bones of ovariectomized animal subjects and is directly correlated with bone resorptive activity. Action of oestrogen is mediated by two isoforms of intracellular oestrogen receptors (ER) receptors, oestrogen receptor-alpha (ERα) and -beta (ERβ), and numerous in vivo studies have demonstrated the essential role of ERα for the regulation of bone metabolism. These intracellular oestrogen receptors (ER) have been described in human periodontal ligament cells. The isoform ERβ is the predominant in the bone metabolism, but both ERα and ERβ isoforms may play a significant role in the osteogenic differentiation of periodontal ligament cells. Theoretically, therefore, withdrawal of oestrogen has been associated with impaired osteogenic differentiation of this cell line. The effect of oestrogen on the oesteogenic differentiation in periodontal ligament stem cells has been examined in rat model of isolated hormone deficiency in the study of Zhang B. et al. [26], six ovariectomized rats were investigated, in comparison with sham-operated rats. When PDCLs were analysed, a significant reduction in expression levels of oestrogen receptors (ERα and ERβ) was observed in PDCLs of ovariectomized rats. These results provided direct evidence for the involvement of oestrogen in the regulation and preservation of osteogenic differentiation of PDCLs, but the underlying mechanism of this process has not been identified. Oestrogen receptors expression has been reported in a variety of cells in bone and bone marrow. Pan F. et al. [27] emphasized the importance of the ER-mediated signalling pathway in the osteogenic differentiation of PDLSCs. Contrasting with previous reports, the author reported an increase of mRNA and protein expression of Erα oestrogen-mediated and a decrease of ERβ in mesenchymal stem cells. Data from three studies [24,25,29] included in the current review also demonstrated severe alveolar bone loss caused by unbalanced bone metabolism, characterized by the high bone resorption activity immediately after ovariectomy and evident alterations in bone formation. Ling-Ling E. et al. [23] examined the effects of oestrogen on the potential of periodontal ligament stem cells (PDLSCs) derived from osteoporotic rats. In this study, the rats were ovariectomized to induce osteoporosis and the PDLSCs derived from the alveolar bone loss were collected. The same procedure was repeated in a sham-operates rats group. PDLSCs were expanded and seeded on a collagen-based composite scaffold [nano-hydroxyapatite/collagen/poly(L-lactide) (nHAC/PLA)] in vitro. Firstly, results showed that the hormone deprivation induced a decreased osteogenic differentiation in OVX rats; then, when the PDLSCs were cultured on nHAC/PLA, it was demonstrated a reduced level of ALP activity, osteocalcin (OCN) secretion, mineral formation and the mRNA expression levels of ALP, OCN, oestrogen receptor (ER)α and ERβ. To assess the effect of oestrogen treatment on the proliferative ability of the cells derived from OVX rats, slides were seeded with 17β-oestradiol (E2). The enhanced osteogenic differentiation was observed, as well as the mRNA expression levels of alkaline phosphatase (ALP), osteocalcin (OCN), Oestrogen receptor (ER)α and ERβ. Data obtained confirmed that the ability of PDLSCs in periodontal regions are affected by a combination of variables, principally reflected in oestrogen deficiency who also described the enhanced ability of PDLCs on the osteogenic differentiation following 17β-oestradiol (E2) treatment in the OVX rats. This is supported by the histological analysis results, which showed pronounced osteogenic differentiation and the mRNA expression levels of alkaline phosphatase (ALP), osteocalcin (OCN), ERα and ERβ ability in the PDLCs isolated from OVX rats after treatment with oestradiol, as revealed by the intense staining for ALP, calcium deposition, and more mineral deposits [31,32,33]. Decreased oestrogen levels again were demonstrated to be directly associated with decreased bone formation in the OVX-PDLSCs group, without treatment with 17β-oestradiol. It was remarkable to note that the study conducted by Ejiri S. et al. [19], extended the analysis of the additional effects of ovariectomy on alveolar bone of rats by considering the effect of the biting force, a form of mechanical occlusal stress. The bone morphometry revealed a significant suppression of bone formation in the alveolar bone around the teeth with a reduced functional mechanical pressure. Summarily, the evidence suggests that the decrease in oestrogen activity leads to an accelerated bone resorptive activity in the alveolar bone, with further accelerating bone resorptive activity at sites with occlusal hypofunction. Liang L. et al. [28] group revealed that in the hPDL-1 and hPDL-2 cells, obtained from periodontal ligament of human teeth (extracted for orthodontic reasons), the expression level of OPG protein was significantly increased and RANKL was downregulated, when the expression of oestrogen receptor beta (Erβ) in hPDL cells was inhibited. Further studies on the effects of oestrogen deficiency, during menopause, on the regulation of bone resorption in periodontium can be found in the work of Liang L. et al. [28] that designed the short interfering RNA technique (siRNAs) targeting different sites of the hPDL cells to inhibit the expression of oestrogen receptor beta (Erb). The primary aim of this study was to investigate how oestrogen affects the expression of OPG and RANKL in human PDL (hPDL) cells. The authors registered an upregulation of OPG expression in hPDL cells and a simultaneous downregulation of RANKL after the stimulating synthesis of oestradiol. These findings support previous results which sustain the potential importance of the oestrogen in regulating the OPG/RANKL system in hPDL cells. These studies investigating the impact of oestrogen on periodontal ligament (PDL) cells, demonstrated that PDL cells synthesize the receptor activator of nuclear factor-kappa B ligand (RANKL) and its decoy receptor, namely, osteoprotegerin (OPG), which directly controls osteoclastogenesis. Whang Y. et al. [25] demonstrated that oestrogen suppression in rat model induces a reduced differentiation of DPSCs towards the osteo/odontogenic cell lineages. In this study it was revealed that the proliferation of DPSCs was markedly reduced compared with physiologically relevant concentrations 107 and 109 M 17b-oestradiol. These data suggested that high concentration of 17b-oestradiol can significantly inhibit proliferation of OR-positive cells. Interestingly, the study of Cai C. et al. [26] focused on the expression of oestrogen receptors ERα and Erβ. They detected increased expression of both receptors. This phenomenon is conflicting with previous reports that indicated that oestrogens increase the mRNA and protein expression of Erα and decrease of ERβ in mesenchymal stem cells [16]. Finally, the results of Ou Q. et al. [22] indicated the ability of E2 to enhance the proliferation and up-regulation stemness-related genes expression. They also demonstrated the increased osteogenic differentiation and elevation of the positive rate of CD146 and STRO-1 over 10 passages in hPDLSCs.

## 5. Conclusions

The present systematic review indicated menopause patients are more susceptible to experimental investigations have provided proof of increased prevalence of periodontal alveolar bone defects in presence of oestrogen deficiency. However, the mechanism by which oestrogen deficiency causes bone loss remains largely unknown. The translational validity of animal models of human menopause it is dubious, because of dissimilarity in bone remodelling conditions. Moreover, rodents do not suffer natural menopause.

## Figures and Tables

**Figure 1 healthcare-09-00572-f001:**
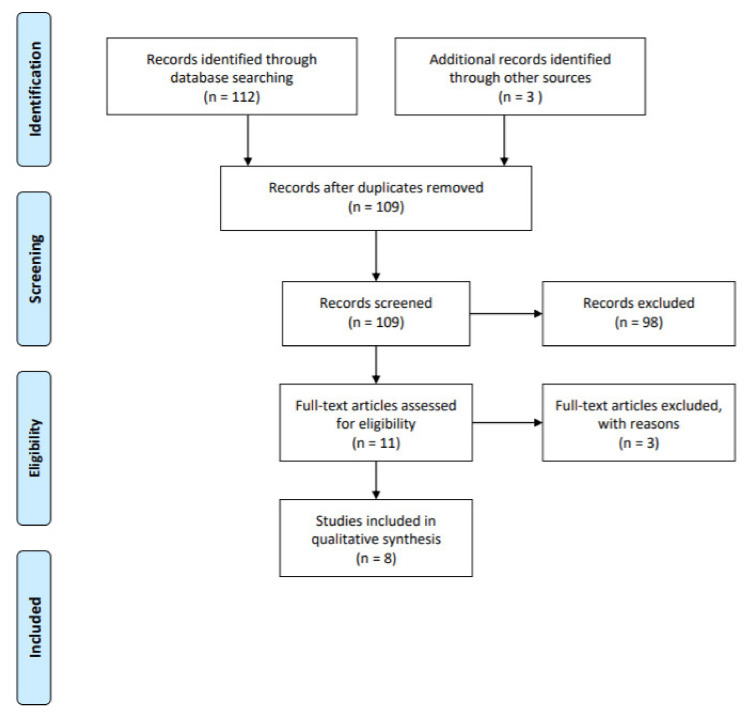
PRISMA flowchart for systematic review and metanalysis.

**Table 1 healthcare-09-00572-t001:** Characteristics of the included studies.

Author and Year of Publication.	Main Characteristics of Models	Cell Lines Origin	Post-OVX Period (months)	Outcomes	Method for the Measurement of Bone Density and/or Oestrogen Deficiency
OU Q. et al., 2018 [22]	12 caries-free premolars extracted for orthodontic reasons from donors aged between 18 and 20	hPDLSCs	NR	Effects of E2 on hPDLSCs	Quantitative real-time reverse transcription polymerase chain reaction (RT-PCR); Western blotting; Immunofluorescence staining analysis
LING-LING E. et al., 2016 [23]	48 healthy Sprague-Dawley female rats were randomly divided into three groups: OVX group; OVX + 17β oestradiol (E2) group, and Sham group	PDLSCs cells of animal origin	3	Effect of oestrogen on osteogenic differentiation in the PLDSCs; The role of oestrogen in regulating the mRNA expression levels of ALP, OCN, ERα and ERβ in the PLDSCs	Immunochemistry assay; Scanning electron microscopy (SEM) analysis; Real-Time reverse transcription polymerase Chain Reaction; Histological and morphometric study
CAI C. et al., 2013 [24]	Periodontal ligament (PDL) tissues were harvested from healthy premolars extracted for orthodontic reasons. Seven donors (12-16 years of age; four females and three males)	hPDLSCs	NR	ALP activity and OCN level of PDLSCs in the presence of oestrogens	Real-time reverse transcription polymerase chain reaction; Immunohistochemical analysis
WANG Y. et al., 2013 [25]	12 caries-free human premolars extracted from six patients due to the orthodontic reasons	human DPSCs	NR	Effects of 17b-oestradiol on odonto/osteogenic differentiation of DPSCs	Real-time reverse transcription polymerase chain reaction
ZHANG B. et al., 2011 [26]	20 Sprague Dawley (SD) rats were randomly divided into two groups: to the OVX group and Sham group.	PDLSCs cells of animal origin	3	The effect of oestrogen depletion on osteogenic differentiation of PDLSCs	Real-Time polymerase Chain Reaction
PAN F. et al., 2011 [27]	Partially impacted third molars (n = 6) were collected from three female individuals aged 18, 19 and 22	human PDLSCs	NR	Effect of oestrogens on the osteogenic differentiation of PDLSCs in vitro	Real-Time polymerase Chain Reaction; Western Blot analysis Immunocytochemical analysis
LIANG L. et al., 2008 [28]	Premolars extracted from 4 premolars Donor’s age: 12 to 14 years old	PDL cells of human origin (hPDL) obtained from the middle of tooth root	NR	The effect of oestrogen depletion on osteogenic differentiation of hPDL cells The effect of oestrogen receptor beta (ERβ) inhibition on osteoblastic differentiation function of human periodontal ligament cells (hPDL)	Real-Time polymerase Chain Reaction; Western Blot analysis
LIANG L. et al., 2008 [29]	Premolars extracted from 4 premolars. Donor’s age: 10 to 12 years old	PDL cells of human origin (hPDL) obtained from the middle of tooth root	NR	The effect of oestrogen on the expression of OPG and RANKL in human PDL (hPDL)	Short interfering RNA technique

**Table 2 healthcare-09-00572-t002:** Risk of bias assessment.

Study	Sample Size Calculation	Random Allocation to Treatment	Blinded Assessment of Outcomes	Compliance With Animal/Human Welfare Regulations	Conflicts of Interest Disclosed	Peer-Reviewed Publication	Quality Score
OU Q. et al., 2018 [22]	N	N	N	Y	N	Y	2
LING-LING E. et al., 2016 [23]	N	Y	N	Y	N	Y	3
CAI C. et al., 2013 [24]	N	Y	N	N	N	Y	2
WANG Y. et al., 2013 [25]	N	Y	N	Y	N	Y	3
ZHANG B. et al., 2011 [26]	N	Y	N	Y	Y	Y	4
PAN F. et al., 2011 [27]	N	Y	N	Y	Y	Y	4
LIANG L. et al., 2008 [28]	N	N	N	N	N	Y	1
LIANG L. et al., 2008 [29]	N	Y	N	Y	N	Y	3

## Data Availability

All data to support the findings of this study are available contacting the corresponding author upon request. The authors have annotated the entire data building process and empirical techniques described in the paper.

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
