# Peer review of "The Effect of Menopause Hypoestrogenism on Osteogenic Differentiation of Periodontal Ligament Cells (PDLC) and Stem Cells (PDLCs): A Systematic Review"

_healthcare, 2021, doi:10.3390/healthcare9050572_

Round 1

Reviewer 1 Report

Dear authors, 

you performed an interesting review about the role of estrogen on Osteogenic Differentiation of Periodontal Ligament cells and Stem Cells (PDLCs). The paper is of interest for dentistry and, in general, for bone specialist. Search method is appropriate and the results are well argued in the discussion. However, you could link your findings with clinical indications and how available interventions, such as dietary supplement, nutraceutic and anabolic drugs could influence periodontal bone health in this population. Please see and cite.

1) Nastri L, Moretti A, Migliaccio S, Paoletta M, Annunziata M, Liguori S, Toro G, Bianco M, Cecoro G, Guida L, Iolascon G. Do Dietary Supplements and Nutraceuticals Have Effects on Dental Implant Osseointegration? A Scoping Review. Nutrients. 2020 Jan 20;12(1):268. doi: 10.3390/nu12010268. PMID: 31968626; PMCID: PMC7019951.  

2) Cecoro G, Paoletta M, Annunziata M, Laino L, Nastri L, Gimigliano F, Liguori S, Toro G, Moretti A, Guida L, Iolascon G. The role of bone anabolic drugs in the management of periodontitis: a scoping review. Eur Cell Mater. 2021 Mar 18;41:316-331. doi: 10.22203/eCM.v041a20. PMID: 33733451.

Author Response

# REVIEWER 1:

Dear authors, 

you performed an interesting review about the role of estrogen on Osteogenic Differentiation of Periodontal Ligament cells and Stem Cells (PDLCs).

  • The paper is of interest for dentistry and, in general, for bone specialist.

Thank’s for your consideration!

  • Search method is appropriate and the results are well argued in the discussion.

Thank’s for your consideration!

  • However, you could link your findings with clinical indications and how available interventions, such as dietary supplement, nutraceutic and anabolic drugs could influence periodontal bone health in this population. Please see and cite.

The following findings have been added:

It has been emphasized that dietary supplement, nutraceutic and anabolic drugs could influence periodontal bone health in this population. It has been established the crucial function of the normal vitamin D status in skeletal mineralization, pointing out on its role in immunity and inflammatory pathway which is revealed through an increased release of anti-inflammatory cytokines and decreasing pro-inflammatory cascade [5-6].

1) Nastri L, Moretti A, Migliaccio S, Paoletta M, Annunziata M, Liguori S, Toro G, Bianco M, Cecoro G, Guida L, Iolascon G. Do Dietary Supplements and Nutraceuticals Have Effects on Dental Implant Osseointegration? A Scoping Review. Nutrients. 2020 Jan 20;12(1):268. doi: 10.3390/nu12010268. PMID: 31968626; PMCID: PMC7019951.  

2) Cecoro G, Paoletta M, Annunziata M, Laino L, Nastri L, Gimigliano F, Liguori S, Toro G, Moretti A, Guida L, Iolascon G. The role of bone anabolic drugs in the management of periodontitis: a scoping review. Eur Cell Mater. 2021 Mar 18;41:316-331. doi: 10.22203/eCM.v041a20. PMID: 33733451.

Reviewer 2 Report

Reviewer comments on Di Naro et al:

Article: The Effect of Menopause Hypoestrogenism on Osteogenic Dif-2 ferentiation of Periodontal Ligament cells (PDLC) and Stem 3 Cells (PDLCs): A Systematic Review.

This review aims to discuss the dynamic correlation between physiological changes of the menopause and the effects on the alveolar bone microenvironment.

Please find below an enumerated list of comments on my review of the manuscript:

INTRODUCTION:

LINE 31: Julie 2020 for July 2020.

LINE 45:

There are some major comments for this section. In the introduction, the manuscript does not provide a complete description of the concept of menopause. It will be useful also to correlate this pheomenon to the age of the woman: the age of a woman is considered a watershed between the fertile and aged stages, so it is important to highlight also this linkage between menopause and aging process (Ultrastructural and morphometric evaluation of aged cumulus-oocyte- complexes- 2013; Ultrastructural markers of quality are impaired in human metaphase II aged oocytes: a comparison between reproductive and in vitro aging -2015).

LINE 53:

The progressive estrogen decline, which characterized the menopause, is also a driving force behind the reproductive aging (RA), a time-dependent deterioration that involves the female germ cells, which occurs in women older than 35 years old (Ultrastructural and morphometric evaluation of aged cumulus-oocyte- complexes- 2013; Ultrastructural markers of quality are impaired in human metaphase II aged oocytes: a comparison between reproductive and in vitro aging -2015). In essence, researchers should highlight also this underlying association, providing a description of the reproductive aging.

LINE 58:

The estrogen decline creates also a room for the activity of anthropogenic substances, characterized by estrogenic properties. Among them, the endocrine disruptor compunds, which compromise bone remodelling process and calcium metabolism (Association between female reproductive health and mancozeb: Systematic review of experimental models – 2020), exerting deleterious and pleiotropic effects on reproductive and bone tissues.

MATERIAL AND METHODS:

As regards this section, the methodology design was rigorous and appropriately implemented within the study.

RESULTS:

Also this section is well organized and densely presented, based on well-synthetized data.

LINE 203:

Also in this section, the paper will benefit from highlighting the correlation between sex steroid hormones decline and the occurence of reproductive aging, in the woman (Ultrastructural and morphometric evaluation of aged cumulus-oocyte- complexes- 2013; Ultrastructural markers of quality are impaired in human metaphase II aged oocytes: a comparison between reproductive and in vitro aging -2015).

Overall, the manuscript requires major changes (as mentioned). I would accept the manuscript, if the comments are addressed properly.

Author Response

# REVIEWER 2:

Reviewer comments on Di Naro et al:

Article: The Effect of Menopause Hypoestrogenism on Osteogenic Dif-2 ferentiation of Periodontal Ligament cells (PDLC) and Stem 3 Cells (PDLCs): A Systematic Review.

This review aims to discuss the dynamic correlation between physiological changes of the menopause and the effects on the alveolar bone microenvironment.

Please find below an enumerated list of comments on my review of the manuscript:

INTRODUCTION:

  • LINE 31: Julie 2020 for July 2020.

It has been corrected with July 2020.

  • LINE 45: There are some major comments for this section. In the introduction, the manuscript does not provide a complete description of the concept of menopause. It will be useful also to correlate this pheomenon to the age of the woman: the age of a woman is considered a watershed between the fertile and aged stages, so it is important to highlight also this linkage between menopause and aging process (Ultrastructural and morphometric evaluation of aged cumulus-oocyte- complexes- 2013; Ultrastructural markers of quality are impaired in human metaphase II aged oocytes: a comparison between reproductive and in vitro aging -2015).
  • The following description has been added: Insufficient estrogens release may have a greater impact on periodontal tissues of women during menopause or already suffering from pre-existing periodontal disease [1-3]. Menopause is the result of loss of ovarian follicular function and coincides with spontaneous permanent cessation of menstrual cycle. This period is typified endocrinologically by the decreasing of ovarian activity with a substantial change occurring in the source and nature of circulating estrogens [1-4].
  • LINE 53: The progressive estrogen decline, which characterized the menopause, is also a driving force behind the reproductive aging (RA), a time-dependent deterioration that involves the female germ cells, which occurs in women older than 35 years old (Ultrastructural and morphometric evaluation of aged cumulus-oocyte- complexes- 2013; Ultrastructural markers of quality are impaired in human metaphase II aged oocytes: a comparison between reproductive and in vitro aging -2015). In essence, researchers should highlight also this underlying association, providing a description of the reproductive aging.
  • The following references have been added:

Bianchi S, Macchiarelli G, Micara G, Aragona C, Maione M, Nottola SA. Ultrastructural and morphometric evaluation of aged cumulus-oocyte- complexes. Italian Journal of Anatomy and Embryology. 2013 Sep;32(9):1343-58. doi: 10.13128/IJAE-13929.

Bianchi S, Macchiarelli G, Micara G, Linari A, Boninsegna C, Aragona C, Rossi G, Cecconi S, Nottola SA. Ultrastructural markers of quality are impaired in human metaphase II aged oocytes: a comparison between reproductive and in vitro aging. J Assist Reprod Genet. 2015 Sep;32(9):1343-58. doi: 10.1007/s10815-015-0552-9. Epub 2015 Aug 15. PMID: 26276431; PMCID: PMC4595403.

  • LINE 58: The estrogen decline creates also a room for the activity of anthropogenic substances, characterized by estrogenic properties. Among them, the endocrine disruptor compunds, which compromise bone remodelling process and calcium metabolism (Association between female reproductive health and mancozeb: Systematic review of experimental models – 2020), exerting deleterious and pleiotropic effects on reproductive and bone tissues.
  • The following description has been added: “Moreover, the characteristics of estrogens deficiency include the induction of anthropogenic substances activity, distinguished by estrogenic properties. Among them, have been found a group of exogenous substance identified as endocrine disruptors to leach endocrine-disrupting chemicals,that interfere with bone remodelling process and calcium metabolism, exerting deleterious and pleiotropic effects on reproductive and bone tissues.”

MATERIAL AND METHODS:

  • As regards this section, the methodology design was rigorous and appropriately implemented within the study.

Thank’s for your consideration!

RESULTS:

  • Also this section is well organized and densely presented, based on well-synthetized data.

Thank’s for your consideration!

LINE 203: Also in this section, the paper will benefit from highlighting the correlation between sex steroid hormones decline and the occurence of reproductive aging, in the woman (Ultrastructural and morphometric evaluation of aged cumulus-oocyte- complexes- 2013; Ultrastructural markers of quality are impaired in human metaphase II aged oocytes: a comparison between reproductive and in vitro aging -2015).

  • The following description has been added:

Insufficient estrogens release may have a greater impact on periodontal tissues of women during menopause or already suffering from pre-existing periodontal disease [1-3]. Menopause is the result of loss of ovarian follicular function and coincides with spontaneous permanent cessation of menstrual cycle. This period is typified endocrinologically by the decreasing of ovarian activity with a substantial change occurring in the source and nature of circulating estrogens [1-4].”

“The progressive estrogen decline, which characterized the menopause, is also a driving force behind the reproductive aging (RA), a time-dependent deterioration that involves the female germ cells, which occurs in women older than 35 years old”.

  • The following references have been added:

Bianchi S, Macchiarelli G, Micara G, Aragona C, Maione M, Nottola SA. Ultrastructural and morphometric evaluation of aged cumulus-oocyte- complexes. Italian Journal of Anatomy and Embryology. 2013 Sep;32(9):1343-58. doi: 10.13128/IJAE-13929.

Bianchi S, Macchiarelli G, Micara G, Linari A, Boninsegna C, Aragona C, Rossi G, Cecconi S, Nottola SA. Ultrastructural markers of quality are impaired in human metaphase II aged oocytes: a comparison between reproductive and in vitro aging. J Assist Reprod Genet. 2015 Sep;32(9):1343-58. doi: 10.1007/s10815-015-0552-9. Epub 2015 Aug 15. PMID: 26276431; PMCID: PMC4595403.

Overall, the manuscript requires major changes (as mentioned). I would accept the manuscript, if the comments are addressed properly.

# Editor comments:

Also, we have noticed that the repetition rate of this manuscript is high. We
have attached the manuscript in this e-mail with the highlighted areas that
need to be revised. Please kindly check and revise them.

The highlighted areas have been revised, as follows:

Line 148-151

  • The electronic searches were supplemented by scanning the reference lists from retrieved articles to identify additional studies that may have been missed during the initial search.

Changed with:

The electronic searches were supplemented by manual searching of reference lists and reviews identify additional primary studies.

Table 1

  • Healthy premolars were extracted from 12 adults (18-20 years of age; 6 male and 6 female) for the reason of orthodontics

Changed with:

12 caries-free premolars extracted for orthodontic reasons from donors aged between 18 and 20

  • Effect of estrogen deficiency PDLSCs cells;

Effect of estrogen on mineral formation in the PLDSCs;

Changed with:

Effect of estrogen deficiency on osteogenic differentiation in the PLDSCs

  • Effect of estrogen on the mRNA expression levels of ALP, OCN, ERα and ERβ in the PLDSCs

Changed with:

The role of estrogens in regulating the mRNA expression levels of ALP, OCN, ERα and ERβ in the PLDSCs

  • Immunochemistry;
  • Scanning electron microscopy;
  • Real-Time polymerase Chain Reaction;
  • Histological and morphometric analysis of the in vivo experiments

Changed with:

Immunochemistry assay;

Scanning electron microscopy (SEM) analysis;

Real-Time reverse transcription polymerase Chain Reaction;

Histological and morphometric study

  • Non-carious human premolars (n = 12) extracted from six young female patients requiring orthodontic treatment, at the age of 12/13

Changed with:

12 caries-free human premolars extracted from six patients due to the orthodontic reasons

Line 204-213

  • …the discrepancy in research methods, the different biological characteristics based on species, sex, and age of the animals across the studies. 2 studies used hPDL cells. The strains used in the studies included ddy, Sprague-Dawley, Swiss-Webster, mice and rats. The age of the rats when the OVX was performed ranged from 4 weeks to 25 weeks. Any study reported the sample size calculation. The sample size of the OVX and control groups in all 7 studies ranged from 12 to 48. The post-OVX period for examination was performed from 3-4 weeks to 3 months after operation

Changed with:

The discrepancy in conducting quantitative research study, the differences in studying the biological characteristics of species across the studies. Two studies investigated hPDL cells. A total of three strains were used in these studiesincluding ddy, Sprague-Dawley, Swiss-Webster, mice and rats.  The OVX was performed about 4 to 25 weeks of age. All studies used post-OVX periods, and in the postoperative period follow up examinations were performed at 3-4 weeks to 3 months. With regard to the sample size, no trials reported any sample size computation. The median sample size of the OVX and control groups in all 7 studies was 23 ± 7

Line 222-224

  • The lowest quality score was 1 [28], and the highest quality score was 4. No study explained the sample size assessment, and no studies used a blinding method in outcome evaluation

Changed with:

Total scores ranged from 1 to 4 (max=4). A quality score of 4 was assigned to 2 studies [26,27]. Only one study was awarded a single point [28], and two awarded two points. Of the all studies, two awarded the highest quality score.  No blinding method was classified.

Line 228-232

  • …based on the effects of osteoporosis. In a previous meta-analysis, meta-regression signaled an association between the post-OVX period and BMD changes in the rats’ mandible

Changed with:

The majority of studies observed the microarchitecture deterioration in the the alveolar bone that characterizes the osteoporosis. The association between the post-OVX period and Bone Mass Density (BMD) changes has been evaluated in several studies. A previous meta-analysis described a more notable decrease in BMD in animals with longer post-OVX periods.

Round 2

Reviewer 2 Report

Manuscript has been extensively  revised and it can be now accepted